# Designing a Multiplex PCR-xMAP Assay for the Detection and Differentiation of African Horse Sickness Virus, Serotypes 1–9

**DOI:** 10.3390/microorganisms12050932

**Published:** 2024-05-03

**Authors:** Martin Ashby, Rebecca Moore, Simon King, Kerry Newbrook, John Flannery, Carrie Batten

**Affiliations:** 1The Pirbright Institute, Ash Road, Pirbright, Woking, Surrey GU24 0NF, UK; rfmoore@live.co.uk (R.M.); simon.king@pirbright.ac.uk (S.K.); kerry.newbrook@pirbright.ac.uk (K.N.); carrie.batten@pirbright.ac.uk (C.B.); 2Department of Pharmaceutical Sciences and Biotechnology, Technological University of the Shannon, Athlone Campus, N37HD68 Athlone, Ireland; johnt.flannery@tus.ie

**Keywords:** AHSV, xMAP, multiplex, serotyping, Luminex^®^, orbivirus

## Abstract

African horse sickness is a severe and often fatal disease affecting all species of equids. The aetiological agent, African horse sickness virus (AHSV), can be differentiated into nine serotypes. The identification of AHSV serotypes is vital for disease management, as this can influence vaccine selection and help trace disease incursion routes. In this study, we report the development and optimisation of a novel, molecular-based assay that utilises multiplex PCR and microsphere-based technology to expedite detection and differentiation of multiple AHSV serotypes in one assay. We demonstrated the ability of this assay to identify all nine AHSV serotypes, with detection limits ranging from 1 to 277 genome copies/µL depending on the AHSV serotype. An evaluation of diagnostic sensitivity and specificity revealed a sensitivity of 88% and specificity of 100%. This method can serotype up to 42 samples per run and can be completed in approximately 4–6 h. It provides a powerful tool to enhance the rapidity and efficiency of AHSV serotype detection, thereby facilitating the generation of epidemiological data that can help understand and control the incidence of AHSV worldwide.

## 1. Introduction

African horse sickness (AHS) is a lethal, economically important, World Organisation for Animal Health (WOAH)-notifiable disease that is caused by the aetiological agent African horse sickness virus (AHSV). AHSV is non-contagious and is transmitted by arthropod vectors, predominantly of the *Culicoides* genus [1]. AHS affects all species of the *Equidae* family, and case fatality rates can approach 90% [2,3]. AHSV is endemic in sub-Saharan Africa, but incursions into both North Africa [4] and non-African countries such as Spain [5] and Thailand [6] have occurred. The virus belongs to the family *Sedoreoviridae*, within the genus *Orbivirus*. Other important viruses within the *Orbivirus* genus include bluetongue virus (BTV), which is the prototype species, and epizootic haemorrhagic disease virus (EHDV). Over the last two decades, BTV has extended its range from southern Europe into northern regions [7], culminating in notable outbreaks in north-western Europe in 2023 [8]. In addition, EHDV was detected in Europe for the first time in 2022 [9]. A causal link between the spread of BTV into new territories and climate change has been suggested [10], driven primarily by the effect of warmer temperatures on vector survival rates and vector competence [11]. Related orbiviruses, such as AHSV, which share the same or similar vectors, also pose a significant risk of incursion into new regions [11].

AHSV has a segmented, double-stranded RNA genome that encodes seven structural (VP) and four non-structural (NS) proteins. The genome is surrounded by a triple-layered capsid comprised of structural proteins VP3 (inner layer), VP7 (middle layer), and VP2 and VP5 (outer layer) [12]. As one of the two outermost proteins of the AHSV capsid, the VP2 spike, which is encoded by genome segment 2 and is comprised of four domains (hub, body, hairpin, and external tip), is a major target of neutralising antibodies. This has driven selective pressure on the encoding gene that has resulted in it being the least conserved of the AHSV proteins [12]. The neutralising antibodies generated against VP2 are serotype-specific. Nine serotypes of AHSV have been identified based on the ability of serotype-specific antibodies to neutralise the virus [12,13].

Identifying the specific AHSV serotype at the onset of an outbreak or during disease surveillance in epizootics is crucial for effective disease control. This includes ensuring optimal vaccine deployment and accurate tracing of incursion routes. While viruses can be serotyped using serological techniques such as the virus neutralisation test (VNT), this technique is highly labour-intensive, requiring the use of the live virus and cell culture facilities, and as some AHSV serotypes are known to cross-react, results can be ambiguous [14]. Molecular techniques offer a more rapid and specific approach to virus typing, and polymerase chain reaction (PCR)-based methods, which amplify specific regions of the VP2 gene that have been found to be unique to each serotype, have been designed for this purpose [15,16,17,18]. However, since the initial design of these assays, a significant amount of sequencing data for AHSV has become available. Therefore, these existing assays need to be re-evaluated to ensure they are capable of detecting currently circulating AHSV strains. Whilst a more recent assay was developed for AHSV serotype differentiation in 2019 [19], it relies on conventional PCR, lacking the sensitivity of real-time PCR and being unsuitable for multiplexing.

Given the multiple AHSV serotypes, it would be highly desirable to develop multiplexing diagnostic molecular assays to enable the detection of multiple (or all) AHSV serotypes within a single reaction. This would substantially reduce time to results as well as reagent costs. However, challenges are frequently encountered when multiplexing RT-qPCR assays, which include loss of sensitivity, tendency for non-specific reactions, and the bleeding of fluorescence into off-target detectors of the PCR instrument. These challenges can considerably complicate assay development and may limit their application. Nevertheless, in 2015, Weyer et al. [16] successfully used the AHSV VP2 gene sequence data that was available at that time to design three triplex RT-qPCR assays capable of detecting and differentiating between the nine AHSV serotypes. Additionally, a more recent triplex RT-qPCR was developed [20] utilizing primers and probes for AHSV serotype differentiation that had been previously published by Duran-Ferrer et al. [21].

It has been demonstrated that xMAP (Multiple Analyte Profiling of an unknown “x”) technology is a suitable platform for the development of multiplex protein- [22], immunological [23,24], and nucleic acid-based assays [25,26], demonstrating potential to aid in the rapid diagnosis of both human and animal pathogens. xMAP methodology for nucleic acid detection (herein referred to as multiplex PCR-xMAP) encompasses several steps, the first being a multiplexed end-point PCR in which nucleic acid is amplified using biotinylated forward or reverse primers. The biotinylated PCR product obtained is then combined with microspheres that have been pre-conjugated with DNA probes specifically designed to hybridise with the serotype-specific PCR product. Fluorescence-based detection is performed using an xMAP instrument (akin to a flow cytometer), distinguishing individual microsphere sets by the fluorescence of their impregnated dye and detecting the presence of a bound analyte (in this case, the serotype-specific PCR product) using an additional reporter dye bound to the biotin label.

The aim of this study was to improve AHSV serotype detection and differentiation through the development of an efficient multiplex PCR-xMAP methodology for AHSV serotypes 1–9. A thorough analysis of the latest AHSV sequencing data was conducted and used to revise primers and probes that would allow for detection of the broadest range of AHSV strains and differentiate them based on serotype (AHSV 1–9). Following an optimisation process, we successfully developed a PCR-xMAP methodology to detect and differentiate all nine AHSV serotypes utilising only two multiplex reactions. The limit of detection ranged from 1 to 277 genome copies/µL, varying based on the specific AHSV serotype.

## 2. Materials and Methods

### 2.1. Virus Panel

This experimental work used AHSV reference strain isolates stored at the WOAH AHSV reference laboratory at The Pirbright Institute, U.K.: RSArah1/03 (serotype 1), RSArah2/03 (serotype 2), RSArah3/03 (serotype 3), SPArah4/03 (serotype 4), RSArah5/03 (serotype 5), RSArah6/03 (Serotype 6), KENrah7/03 (serotype 7), RSArah8/03 (serotype 8), and PAKrah9/03 (serotype 9). In addition, a panel of eighty-seven archived EDTA blood samples from suspected AHSV-positive field cases and samples from AHSV proficiency testing (PT) schemes (obtained from the Laboratorio Central de Veterinaria, Spain) were used to evaluate diagnostic sensitivity and specificity.

### 2.2. In Silico Screen of AHSV Primers and Probes

An in silico analysis was performed using SCREENED, described by Vanneste et al. [27], to predict the suitability of the primer and probe sequences from previously published AHSV serotyping RT-qPCR assays: Bachanek-Bankowska et al. [15] and Weyer et al. [16]. AHSV VP2 nucleotide sequences were downloaded from GenBank (https://www.ncbi.nlm.nih.gov/genbank/, accessed on the 20 October 2020) using either “African horse sickness virus segment 2” or “African horse sickness virus VP2” as the search terms. The two files were then combined and manually inspected. Duplicate sequences and sequences from a different genome segment or different organisms/viruses were removed from the alignment. In total, 301 AHSV VP2 gene sequences were collected, and the FASTA file containing AHSV segment 2 sequences was used as the input file for SCREENED analysis. Sequences were removed from the FASTA file if any of the primer binding sites for either assay were not present in the available sequence due to incomplete sequencing data. This resulted in the evaluation of 68 sequences for AHSV-1, 16 for AHSV-2, 17 for AHSV-3, 28 for AHSV-4, 15 for AHSV-5, 20 for AHSV-6, 17 for AHSV-7, 23 for AHSV-8, and 20 for AHSV-9.

SCREENED assigns each forward/reverse primer and probe two values to quantify the likelihood of successful amplification and therefore detection of the sequence by RT-qPCR. The first value (v1) describes whether the primer/probe is expected to successfully anneal: an initial value of 1 is given for each primer/probe, followed by a penalty of −4 if there is a mismatch within five bases of the 3′ end (which is associated with a high frequency of primer failure); a penalty of −2 is given if there are more than 20% mismatches in the primers or probe. The second value (v2) is the number of mismatches between the primer/probe and annealing site. We assigned a cumulative score for each set of primers (forward and reverse) and probes from the Bachanek-Bankowska et al. [15] and Weyer et al. [16] assays. An illustrative example of the approach taken is provided below:

*An assessment of the Weyer et al.* [16] *forward and reverse primers and probe for an AHSV-6 strain (GenBank reference KT030411.1) revealed one mismatch between the forward primer and annealing site, giving a v2 score of −1 to account for the mismatch; however, as the mismatch is not within 5 bp of the 3′ end, it was assigned a v1 score of 1; hence, the total score (v1 + v2) for the forward AHSV-6 primer would be 0. The reverse primer and the probe were both found to be exact matches to the target annealing sites, so both received a total score (v1 + v2) of 1. Therefore, the overall cumulative score for the Weyer et al. serotype 6 primers/probe against the AHSV strain KT030411.1 GenBank sequence would be 2 (i.e., 0 + 1 + 1).*

Using this method, the probe and primer (forward/reverse) set can receive a maximum score of three when there are no mismatches within the target annealing site. An average score was generated for each assay against each AHSV serotype. In cases where both assays scored poorly for a serotype, the annealing sites of primers and probes were examined to determine common mismatches present across the segment 2 sequences and whether incorporation of degenerate bases in the primer/probe could improve annealing. These modified sequences were subsequently analysed in SCREENED using the same parameters as above to determine if there was an improvement in the score.

In primers where degenerate bases were incorporated, the absence of cross-reactivity with pathogens causing AHSV differential diseases (equine infectious anaemia virus, Hendra virus, equine encephalosis virus, and *Bacillus anthracis*) was determined using BLAST software (https://blast.ncbi.nlm.nih.gov/, accessed on the 31 August 2023).

The software tool Autodimer (version 1) [28] was then employed to detect potential cross-reactivity among the primers chosen for multiplexing following the SCREENED analysis. Autodimer utilises a sliding algorithm to measure complementarity between DNA oligomers, assigning a score to oligomer pairs to quantify the potential for cross-reactivity. Utilising this tool, the primers for each AHSV serotype were systematically evaluated in conjunction with one another, leading to the design of two multiplex reactions aimed at minimising cross-reactivity (Table 1).

### 2.3. Nucleic Acid Extraction

Nucleic acid extraction was performed using the MagMAX™ CORE Nucleic Acid Purification Kit (Thermo Fisher Scientific, Paisley, UK) and KingFisher™ Flex Purification System using the MagMAX core flex no-heat protocol. For AHSV reference strains, the extraction was performed on 200 µL cell culture virus isolate. For field samples and samples from PT panels, the extraction was performed on 200 µL EDTA blood. In all cases, nucleic acid was eluted into a final volume of 90 µL and stored at +4 °C (if used immediately) or −20 °C (long-term storage).

### 2.4. RT-qPCR

Results obtained using the multiplex PCR-xMAP were compared to those obtained using RT-qPCR. Pan-AHSV RT-qPCR detection was performed using primers and probe published in Agüero et al. [29], targeting a conserved region within AHSV segment 7.

Extracted nucleic acid was denatured at 95 °C for 5 min in a 96-well PCR plate (4 µL extracted nucleic acid per well). Mastermix was prepared from the EXPRESS One-Step Superscript™ qRT-PCR Kit (Thermo Fisher Scientific) using 10 µL EXPRESS SuperScript^®^, 1 µM forward primer, 1 µM reverse primer, 0.25 µM probe, 0.4 µL ROX, and 2 µL EXPRESS SuperScript^®^. The mastermix was added to the 4 µL denatured nucleic acid, resulting in a total reaction volume of 20 µL. Cycling and detection was performed using the 7500 Fast Real-Time qRT-PCR System. Cycling conditions were: 50 °C for 15 min and 95 °C for 20 s, followed by 45 cycles of 95 °C for 3 s, 55 °C for 30 s, and 72 °C for 30 s. C_T_ values less than 45 were considered positive. For quantification, an ultramer was purchased from Integrated DNA Technologies, Belgium, which corresponded to bp 1009 to 1112 of AHSV segment 7 (GenBank reference OP508204.1). This highly conserved region encompasses the primer/probe hybridisation sites for the Agüero et al. assay [29]. The ultramer was diluted ten-fold in molecular-grade water, and dilutions were tested in triplicate in each RT-qPCR assay to enable the creation of a standard curve for quantification of RNA copies in the limit of detection (LOD) studies.

### 2.5. RT-qPCR Serotype Determination

For serotype-specific RT-qPCR, the primers and probes described in Bachanek-Bankowska et al. [15] were used. Extracted nucleic acid was denatured at 95 °C for 5 min in a 96-well PCR plate (3 µL extracted nucleic acid per well). Mastermix was prepared from the EXPRESS One-Step Superscript™ qRT-PCR Kit (Thermo Fisher Scientific, Paisley, UK) using 10 µL EXPRESS SuperScript^®^, 0.8 µM forward primer, 0.8 µM reverse primer, 0.1 µM probe, 0.4 µL ROX, 2 µL EXPRESS SuperScript^®^, and 2.6 µL RNase free water. The mastermix was added to the 3 µL denatured nucleic acid, resulting in a total reaction volume of 20 µL. Cycling and detection was performed using the 7500 Fast Real-Time qRT-PCR System. Cycling conditions were: 50 °C for 15 min and 95 °C for 20 s and 45 cycles of 95 °C for 3 s and 60 °C for 30 s. C_T_ values less than 45 were considered positive.

### 2.6. Preparation of Microspheres

Serotype-specific DNA capture probes (Eurofins genomics, Ebersberg, Germany) were purchased based on previously published RT-qPCR assays (Table 1). Each capture probe was modified to contain a C12 linker with 5′ amino group. Based on the in silico analysis, degenerate bases were incorporated into probes for AHSV-5, -6, and -9 to improve detection. MagPlex microsphere sets 12–22, (Luminex Corporation, Hertogenbosch, The Netherlands) were coupled to DNA capture probes following methods previously described [30]. Each microsphere set was hybridised to capture probes for a distinct AHSV serotype. Briefly, 10^6^ microspheres were suspended in 10 µL 0.1 M 4-Morpholineethanesulfonic acid (MES hydrate), pH 4.5 (Sigma Aldrich, Dorset, UK), and vortexed for 30 s, followed by the addition of 1 µL 50 µM capture probe, 1 µL 10 mg/mL 1,3-Propanediamine,*N*-(ethylcarbonimidoyl)-*N*,*N*-dimethyl-, monohydrochloride (EDC) solution (Thermo Fisher Scientific), then additional vortexing and incubation in the dark (at room temperature) for 30 min. An additional 1 µL 10 mg/mL EDC was added, followed by vortexing and incubation in the dark (at room temperature) for 30 min. Next, 500 µL 0.02% (*v*/*v*) tween-20 (Sigma-Aldrich) was added, followed by vortexing and magnetic separation of the microspheres from the solution. The supernatant was discarded. Microspheres were then resuspended in 500 µL 0.1% Sodium dodecyl sulphate (SDS) (Sigma-Aldrich) and vortexed. Microspheres were separated from solution using a magnet, and the supernatant discarded. Finally, the pellet was resuspended in 200 µL of Tris-EDTA buffer (pH 8) and stored in the dark at 4 °C (±2 °C) until used.

### 2.7. Multiplex PCR-xMAP Amplification and Labelling of Template RNA

Primer sequences used for the amplification of RNA in the multiplex PCR-xMAP were chosen based on in silico analysis of primers published in the scientific literature [15,16]. A list of the primers used in this study is provided in Table 1; note that some primers were modified from that described in the literature source to incorporate degenerate bases. For each sample, 5 µL of the extracted nucleic acid was added to two separate reaction wells (Figure 1i,ii). One reaction well contained 2 μL of multiplex reaction A (4-plex consisting of primers for AHSV-1, -3, -5, and -7), while the other contained 2 μL multiplex reaction B (5-plex consisting of primers for AHSV-2, -4, -6, -8, and -9). Primer concentrations within each multiplex reaction were chosen following careful optimisation (Table 1). The primer mix and extracted nucleic acid were heated to 95 °C for 5 min, followed by rapid cooling to 4 °C. Subsequently, a mastermix was prepared using the EXPRESS One-Step Superscript™ qRT-PCR Kit (ThermoFisher, Paisley, UK), using 10 µL EXPRESS SuperMix Universal, 2 μL of EXPRESS SuperScript, and 1 μL RNase free water. This mastermix was added to each well (Figure 1iii). Amplification was performed in a thermocycler using touch-down PCR, with cycling conditions as described in Table 2 (Figure 1iv).

### 2.8. xMAP Detection of Serotype Specific Amplicons

Microspheres (coupled with capture probes) were diluted in 1.5× TMAC hybridisation solution (4.5 M teramethylammonium chloride solution, 0.15% N-lauroylsarcosine salt solution, 75 mM Tris, 6 mM EDTA, and topped up to the required volume with water), so the solution contained 76 microspheres/μL of each microsphere set. Then, 33 μL of this solution was added to allocated wells of a 96-well plate. For each sample to be tested, two wells of a 96-well plate were required: one containing microspheres with conjugated capture probes for the detection of AHSV-1, -3, -5, and -7 and the other containing microspheres with conjugated capture probes for the detection of AHSV-2, -4, -6, -8, and -9. 5 μL of the amplification product was added to wells containing microspheres, and the wells were topped up to 50 μL with Tris-EDTA buffer (Figure 1v). The plate was covered with a plate seal, and amplicon/microsphere hybridisation was performed by heating the plate to 96 °C for 90 s, followed by 50 °C for 20 min in a thermocycler. Streptavidin, R-Phycoerythrin conjugate (SAPE) was diluted in 1× TMAC (3 M teramethylammonium chloride solution, 0.10% N-lauroylsarcosine salt solution, 50 mM Tris, 4 mM EDTA, and topped up with the required volume of water to 10 µg/mL), and 25 μL was added to each well (Figure 1vi). The plate was then placed in a Bio-Plex 200 instrument (Luminex corp., Hertogenbosch, The Netherlands) set to 50 °C, where it was incubated for 20 min prior to the detection of hybridised amplification product (Figure 1vii). Data were collected using the Luminex^®^ xPONENT 3.1 software. Results were based on detection of median fluorescent intensity unit (MFI) averages across a minimum of 50 microspheres per set, with the blank readings from two negative controls subtracted. Each sample was run in duplicate. A cut-off of 200 MFI was required for a sample to be considered positive. If two AHSV serotypes were detected in a single sample, then the cut-off was increased to 400 MFI before the sample was considered positive for either serotype. This additional cut-off was necessary to negate low-level cross-reactivity between some AHSV serotypes.

## 3. Results

### 3.1. In Silico Screening of Primers and Probes

The selection of forward/reverse primers and capture probes used in the multiplex PCR-xMAP assay was guided by in silico screening. We screened the primers and probes that were described in the Weyer et al. [16] and Bachanek-Bankowska et al. [15] AHSV RT-qPCR assays. Our aim was to predict the primer pairs and probes that would amplify the broadest range of strains within each specific AHSV serotype. To do so, it was important to consider the number and location of any mismatches between primers/probe and the target sequence. The likelihood of successful amplification was calculated and represented using a scoring system. The average score for all strains within an AHSV serotype, when using primers and probes from either the Weyer et al. [16] or Bachanek-Bankowska et al. [15] assays, is provided in Table 3. Our analysis suggested that improvements to some forward/reverse primers and probes could be made through the addition of degenerate bases. When these modified primers were subsequently included in the in silico screen, they obtained a higher score than their unmodified counterparts. The primers/probes that obtained the highest score in the in silico screen were modified versions of the Weyer et al. [16] primers/probes for serotype AHSV-1, -4, -6, and -8 and modified versions of the Bachanek-Bankowska et al. [15] primers/probes for serotypes AHSV-2, -5, and -9. For serotypes AHSV-3 and -7, the primers/probes published by Bachanek-Bankowska et al. [15] without modification scored highest. A BLAST of the primers that were modified with degenerate bases revealed no notable increase in cross-reactivity with AHSV differential diseases.

### 3.2. Multiplex PCR-xMAP Optimisation

A series of different assay parameters were assessed using a range of conditions to monitor their effect on assay performance. Based on the results of these optimization experiments, we decided upon the following approaches:Including the primers in the reaction well when denaturing the double-stranded template RNA at 95 °C prior to addition of mastermix and PCR;Utilising a touchdown PCR method to limit non-specific amplification;Switching from a single 9-plex assay to performing a two-reaction well (4-plex and 5-plex) assay for the detection of all AHSV serotypes;Employing an asymmetric concentration of PCR forward and reverse primers for detection of AHSV-1, -3, -5, -6, -8, and -9 (this biases the amplification towards an increased yield of biotinylated amplicon but did not improve detection of AHSV-2, -4, or -7);Removal of 1.5× TMAC following the hybridisation of microsphere and amplicon, prior to the addition of SAPE. This method, described in Angeloni et al., 2018, titled “direct DNA hybridisation washed protocol”, was found to perform better than the “direct DNA hybridisation: no wash protocol” from the same publication [30];Conducting the microsphere-amplicon hybridisation and the amplicon-SAPE labelling step at 50 °C, which performed better than other higher temperatures that were also tested.

These optimisation steps collectively contributed to improved detection of the target nucleic acid and were thus incorporated into the assay method. Results of the optimisation experiments can be found in the Appendix A.

### 3.3. Detection of AHSV Reference Strains Using the Multiplex PCR-xMAP Assay

To validate the newly developed and optimised multiplex PCR-xMAP assay, we used the assay to detect the AHSV reference strains (AHSV-1 to -9). The correct serotype was identified in all nine AHSV reference strains (Figure 2). The MFI for the target serotypes were on average 100 times greater than the averaged MFI for the off-target serotypes. (The average MFI of off-target serotypes was 39.7, and the average MFI of the target serotype was 4425.6)

### 3.4. Limit of Detection (LOD)

A ten-fold serial dilution was performed on extracted nucleic acid from each of the AHSV reference strains (AHSV-1 to -9). Each sample was tested using both the multiplex PCR-xMAP methodology and the pan-AHSV RT-qPCR assay (Table 4). The multiplex PCR-xMAP MFI and PCR C_T_ value of both assays showed a strong correlation with the RNA copy number, although the relationship was marginally stronger between RNA copy number and C_T_ value (average Pearson r value for log RNA copy number vs. PCR C_T_ = −0.99, log RNA copy number vs. xMAP MFI = 0.95). The multiplex PCR-xMAP MFI values in the undiluted samples ranged from 1618–6516 (average 3773), and corresponding RT-qPCR C_T_ values for these samples ranged from 22.20–24.52. The LOD varied depending on AHSV serotype but ranged from 277 genome copies/µL for AHSV-1 to 1 genome copy/µL for AHSV-9. The average LOD was 29 genome copies/µL. The corresponding C_T_ value for the pan-AHSV RT-qPCR assay, in the highest dilutions at which the AHSV serotype could be detected in the multiplex PCR-xMAP assay, ranged from 31.88–38.74 (average 34.72).

### 3.5. Diagnostic Sensitivity and Specificity of Multiplex PCR-xMAP Assay

Eighty-seven archived equine EDTA blood samples previously submitted to the WOAH AHSV Reference laboratory at The Pirbright Institute (U.K.) for AHSV testing were screened using the optimised AHSV multiplex PCR-xMAP assay to evaluate its diagnostic sensitivity and specificity (summarised in Table 5, with complete data in Appendix A). The results were compared to the WOAH recommended pan-AHSV RT-qPCR assay [29]. The pan-AHSV RT-qPCR assay identified 67/87 samples as positive for AHSV genetic material. A serotype could be detected in 59/67 of these using the multiplex PCR-xMAP assay, resulting in a diagnostic sensitivity of 88% (95% CI = 78% to 94%). Of the eight samples that were pan-AHSV RT-qPCR positive but for which no AHSV serotype was detected using the multiplex PCR-xMAP, three samples had C_T_ values of >38 in the pan-AHSV RT-qPCR and may be beyond the LOD of the multiplex PCR-xMAP. The other five samples that were not detected in the multiplex PCR-xMAP had C_T_ values in the pan-AHSV RT-qPCR that ranged from 22.07–28.28.

The pan-AHSV RT-qPCR identified 20/87 samples as being negative for AHSV genome, and all of these were also found to be negative using the multiplex PCR-xMAP assay, resulting in a diagnostic specificity of 100% (95% CI = 83% to 100%).

Using the multiplex PCR-xMAP assay, six different AHSV serotypes were identified in the panel of blood EDTA samples; these were AHSV-1, -2, -4, -5, -6, and -9. Each AHSV serotype detected in an EDTA blood sample was confirmed using the serotype-specific AHSV RT-qPCR described in Bachanek-Bankowska et al. [15] or, in the case of samples 50–59, by consulting the results provided by the supplier of the PT panel (full data in Appendix A).

## 4. Discussion

The segmented nature of the orbivirus genome allows for the re-assortment of AHSV genome segments between different AHSV strains during viral co-infection of the same cell [31]. This feature, coupled with the error prone replication of RNA viruses, increases genetic diversity and accelerates the ability of orbiviruses to overcome host defences [32]. A total of 22 species of *Orbivirus* genus exist, within which there are at least 160 different serotypes [33]. Within the AHSV species, there are nine recognised serotypes. These can be identified and distinguished using a VNT, an assay based on the detection of serotype-specific neutralising antibodies that are generated against the outermost proteins of the AHSV particle, VP2 and VP5, in response to infection. However, VNTs have several disadvantages. Obtaining results may take up to 7 days, and they require SAPO3 virus-handling infrastructure and cell culture facilities. Additionally, complications arising from cross-reactivity between serotypes can make the interpretation of results particularly challenging. As an alternative to VNTs, molecular-based methods, including conventional RT-PCR [17,18,19] and RT-qPCR [15,16,20], have been designed for the precise identification and differentiation of AHSV serotypes by targeting specific regions within the VP2 gene, which are unique to each serotype. These PCR-based methods provide significantly faster time to results compared to serological methods and are beneficial for their ease of use and high sensitivity.

RT-qPCR methods for AHSV serotyping have been implemented using a single-plex approach, where each reaction is designed to detect a single AHSV serotype only [15,19], or as a multiplex, where a single reaction well has multiple primers and probes that allow for the detection of more than one AHSV serotype simultaneously [16,20]. Multiplexing is especially desirable where large numbers of samples need to be tested, as the number of reaction wells required in a single-plex assay increases exponentially with the number of samples. However, when used as a platform for multiplexing, RT-qPCR has some limitations. Specifically, it is constrained by the limited number of spectrally distinct fluorophores available to uniquely identify targets. Additionally, fluorescence bleeding between detection channels (cross-talk) can lead to incorrect interpretation of results. Therefore, technologies that can combine the speed and accuracy of PCR with the efficiency of reliable multiplexing should be sought. One possible solution to this problem is the use of the xMAP platform developed by Luminex^®^. This technology allows for the simultaneous detection of up to 500 different targets using a method that is more robust to the effects of cross-talk between channels. Various multiplex molecular assays have been developed using the xMAP platform to differentiate between bacteria, fungi, viruses, or genotypes [25,26,34,35,36].

To expedite the development of a PCR-xMAP assay for AHSV serotyping, we first assessed primers and probes from pre-existing RT-qPCR assays to determine their suitability for integration into the assay. Two existing RT-qPCR assays that are commonly used for AHSV serotype identification were published between 2014–2015 [15,16]. However, since then, approximately 250 new submissions of AHSV segment 2 sequencing data have been made publicly available, representing 72% of the total submissions. Using an in silico approach, we were able to map primer/probe binding sites onto the latest VP2 sequencing data to identify positions where the mismatches between primer/probes and their RNA targets occur. It should be noted that while efforts were made to remove irrelevant VP2 sequences from our dataset, in some cases, closely related sequences from a common source were still included. In future work where a similar in silico approach is undertaken, researchers need to carefully consider how including multiple sequences from a single laboratory or outbreak may mean particular mismatches could be overrepresented in the analysis.

As demonstrated in our in silico analysis, several of the primer/probe sets were predicted to perform poorly against numerous AHSV strains/serotypes, demonstrating the need to periodically assess whether primers and probes require re-designing as new AHSV strains emerge. It may also point to limitations in the two RT-qPCR assays [15,16], which are commonly used for AHSV serotyping. Following careful data analysis, we incorporated degenerate basis into several primers/probes. This improved the predicted in silico scores and formed the bases for the design of the multiplex PCR-xMAP assay. It is worth noting a recently developed RT-qPCR assay for AHSV serotype determination was published in 2024 [20] that may offer improved performance against recently sequenced strains. However, due to the recentness of this publication, we were unable to incorporate the primers and probes from this assay into our in silico analysis.

When designing the multiplex PCR-xMAP we initially aimed to combine all primer pairs into a single 9-plex reaction mix. However, despite optimisation, we could not robustly detect all AHSV serotypes using this approach. This is likely to result from the formation of primer-dimers during PCR amplification, a common pitfall of multiplex PCR reactions. To address this issue, we divided the assay into two multiplex reactions, one comprising four primer pairs that are specific for AHSV serotypes -1, -3, -5, and -7 and the other comprised of five primer pairs specific for AHSV serotypes -2, -4, -6, -8, and -9. Utilising online primer dimer prediction tools such as Autodimer [28] along with empirical observation, we designed each multiplex reaction to keep primers that were likely to cross-react separate.

The sensitivity of the xMAP-PCR assay was demonstrated for each AHSV serotype, with limit of detections ranging from 1 to 277 genome copies/µL depending on the serotype tested. The single-plex RT-qPCR was either equally sensitive or more sensitive than the PCR-xMAP assay (dependent on serotype), as it could occasionally detect a positive signal at a dilution that was not always detected in the multiplex. However, this was not unexpected and is probably due to non-specific interactions between primers that are difficult to completely avoid when multiplexing. A decrease in sensitivity of PCR-microsphere multiplex assays compared to RT-qPCR has been reported by others [37]. Nevertheless, the multiplex PCR-xMAP assay successfully identified AHSV serotypes in samples with typical C_T_ values observed in animals during AHSV outbreaks [6].

The clinical specificity and sensitivity of the PCR-xMAP assay was evaluated using 87 samples of equine origin. In all samples that were confirmed as AHSV-negative in the RT-qPCR assay, no AHSV serotype was detected using the multiplex PCR-xMAP, resulting in 100% specificity. The multiplex PCR-xMAP assay was able to identify an AHSV serotype in both field and PT samples, which included a range of AHSV serotypes at a range of C_T_ values. Our results indicated a diagnostic sensitivity of 88.06%. In three AHSV PCR-positive samples, where a serotype could not be detected using the multiplex PCR-xMAP assay, the corresponding C_T_ value in the single-plex RT-qPCR was >35, indicating low levels of viral RNA in these samples. Thus, these three false negatives may be attributed to the inferior LOD of the multiplex PCR-xMAP assay compared to the single-plex RT-qPCR. This explanation cannot be used for the five other false-negative samples, however, for which the C_T_ values using the single-plex RT-qPCR were found to be <30. Due to limited sample availability, it was not possible to carry out follow-up testing on these samples, and therefore, operator error could not be ruled out. Nor could we rule out primer/probe mismatches with cDNA binding sites for these specific strains. Sequencing these samples in the future would be interesting to eliminate that possibility. It is worth noting that whilst the false-negative results occurred in samples containing AHSV-1, -4, -5, and -6, these serotypes were successfully detected in other samples, indicating that there was not a failure associated with a specific AHSV serotype.

A more comprehensive validation of the PCR-xMAP assay should be sought in the future. This can be achieved by encouraging additional laboratories to perform the assay on well-characterised samples, thereby evaluating inter-lab reproducibility. Furthermore, refining estimates of the assay’s sensitivity can be accomplished by testing additional samples from other AHSV reference laboratories and those collected within surveillance programs conducted in settings where AHSV is circulating.

Compared to performing single-plex RT-qPCR reactions, the multiplex PCR-xMAP offers advantages in terms of efficiency and reagent usage. Particularly, in scenarios where serotyping needs to be performed on a large number of samples, the multiplex approach brings about significant savings in both time and cost. For instance, in this study, we successfully identified a serotype in 59 AHSV-positive samples using only 118 PCR-xMAP reactions, which theoretically translates to less than two 96-well PCR plates. In contrast, the single-plex approach would require up to 531 RT-qPCR reactions, requiring over five 96-well PCR plates. The entire PCR-xMAP assay can be performed in approximately 4–6 h (excluding sample extraction), and a single run can serotype up to forty-five samples simultaneously.

Access to tools such as the PCR-xMAP for AHSV serotyping will increase the feasibility of conducting large-scale serotyping studies that can aid in defining the epidemiological landscape of AHS and may quickly elucidate routes of disease incursion from which appropriate risk management strategies can be employed.

## Figures and Tables

**Figure 1 microorganisms-12-00932-f001:**
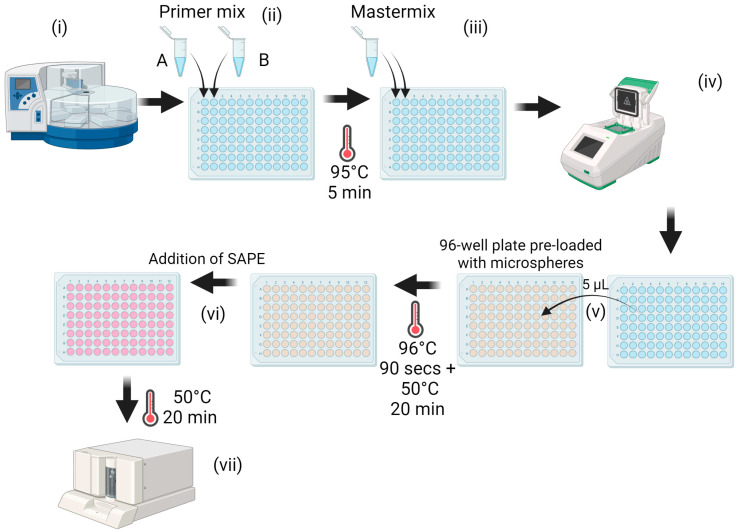
Schematic representation of multiplex PCR-xMAP workflow. Nucleic acid is extracted in an automated extraction robot (i). The resulting eluent is then split across two wells and combined with biotinylated primers from multiplex reaction mix A in one well (for the detection of AHSV serotypes -1, -3, -5, and -7) and multiplex reaction mix B in the second well (for the detection of AHSV serotypes -2, -4, -6, -8, and -9) (ii). This mixture is then denatured, followed by the addition of qPCR mastermix (iii) and then RT-PCR (iv). The resulting amplified product is added to a 96-well plate preloaded with microspheres (v). Hybridisation of amplified product (if present) to microspheres occurs during an incubation step at 50 °C. Streptavidin, R-Phycoerythrin Conjugate (SAPE) is then added to each well and binds to the amplified product (if present) that is attached to the microsphere (vi). Finally, detection of both SAPE and microsphere is performed using the Bio-Plex 200 instrument (vii). Schematic created with Biorender.com.

**Figure 2 microorganisms-12-00932-f002:**
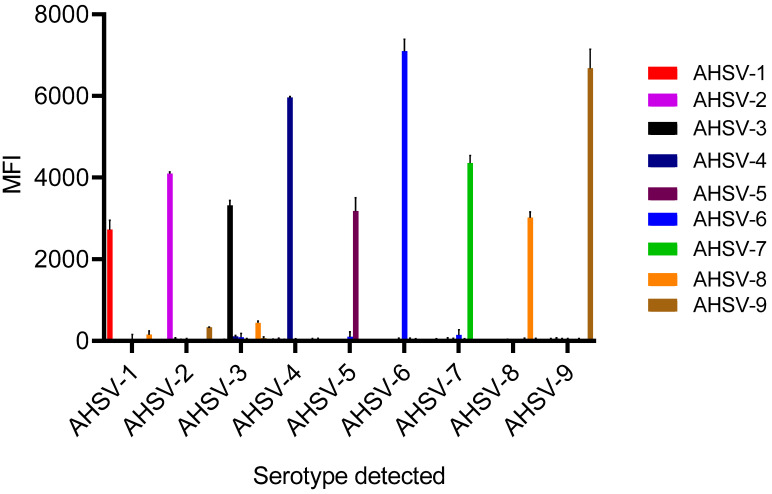
The nine AHSV reference strains for serotypes 1–9 (indicated by different coloured bars) detected using the multiplex PCR-xMAP methodology. The median fluorescent intensity (MFI) is calculated from the average (mean) of each sample run in duplicate. Error bars represent standard deviation values. Table of all values available in Appendix A.

**Table 1 microorganisms-12-00932-t001:** Forward and reverse primers and capture probes used in the development of the multiplex PCR-xMAP assay. The assay was split into two multiplex reactions: A and B.

	AHSV Serotype	Primer	Primer Sequence (5′-3′)	Final Primer Concentration in Mastermix (µM)	Capture Probe Sequence (5′-3′) ^1^	Source
Multiplex reaction A	1	Forward	TGAACATAAACAAAC**R**GTGAGTGA	0.1	CAGTTGAAAAAGAAACAAG	Weyer et al., 2015 [16] ^#^
		Reverse	Btn-GGTTAGAGGYGCTCG**R**TTCT	0.4		
	3	Forward	TAGAAAGAATGATGAGCAGTG	0.2	CCGTTATTGAGAGCGTCATAAGATTC	Bachanek-Bankowska et al., 2014 [15]
		Reverse	Btn-TAATGGAATGTCGCCTGTCTT	0.8		
	5	Forward	GAGACACATCAAGGTTAA**R**GG	0.4	TTGAAGCAAGG**R**ATCTATT**R**ACTTT	Bachanek-Bankowska et al., 2014 [15] ^#^
		Reverse	Btn-CAGGATCAAACTGTGTATACTT	1.6		
	7	Forward	TGGATCGAGCATAAGAAGAAG	0.2	ACCAAAATCGTCCGATGCTAGTGC	Bachanek-Bankowska et al., 2014 [15]
		Reverse	Btn-CCAATCAACCCARTGTGTAAC	0.2		
Multiplex reaction B	2	Forward	Btn-AGTGGACTTCGAT**Y**ATAGATG	0.1	TTCAACCGTCTCTCCGCCTCTC	Bachanek-Bankowska et al., 2014 [15] ^#^
		Reverse	CTGTCTGAGCGTTAACCTC	0.1		
	4	Forward	CATATAAAGGAGGTAACCGARAAAYTG	0.8	AGAAAGCGCAAACCG	Weyer et al., 2015 [16] ^#^
		Reverse	Btn-GGCATGGTTGTCCTCCATTT	0.8		
	6	Forward	TTAATCCGAA**Y**CACCA**R**ACG	0.2	TGATCA**R**ATGAATCGTGCGC	Weyer et al., 2015 [16] ^#^
		Reverse	Btn-GAGGTTTATTA**Y**TGTTGCCTTGC	0.8		
	8	Forward	ACGGCGA**R**AA**Y**TGGAAAAAA	0.2	ATAAGGCGGAAGTCC	Weyer et al., 2015 [16] ^#^
		Reverse	Btn-TGCGCTTCATTCAAACGTT**Y**T	0.8		
	9	Forward	Btn-TATCATATTGGTATCGAGTTCG	0.8	ACA**Y**CCTCAATCGAYCTCCTCTC	Bachanek-Bankowska et al., 2014 [15] ^#^
		Reverse	AAGTTGATGCGTGAATACCGA	0.2		

Btn = primer is biotinylated. ^1^ Capture probes contain 5′ amino group with C12 linker to facilitate hybridisation to microspheres ^#^ The original sequence described in the publication has been modified by incorporation of degenerate bases (emboldened).

**Table 2 microorganisms-12-00932-t002:** Cycling conditions of touch-down PCR used in the multiplex PCR-xMAP method.

Number of Cycles	Temperature (°C)	Time
1	50	15 min
	95	20 s
10	95	3 s
	65–55 (reducing by one degree per cycle)	30 s
	72	30 s
35	95	3 s
	55	30 s
	72	30 s

**Table 3 microorganisms-12-00932-t003:** The average in silico score determined by SCREENED for published forward/reverse primers and probes against strains of different AHSV serotypes. A higher score predicts better performance (maximum 3). The column “modified” represents the in silico score of the published primer and probe following modification by inclusion of degenerate bases.

AHSV Serotype	Number of Strains Included in Screen	In Silico Score Weyer et al. [16]	In Silico Score Bachanek-Bankowska et al. [15]	In Silico Score Modified
1	68	2.62	1.95	**2.95 ^W^**
2	16	2.12	2.81	**3.00 ^B^**
3	17	1.05	**3.00**	N/A
4	28	−2.14	2.11	**2.29 ^W^**
5	15	−3.25	0.66	**3.00 ^B^**
6	20	1.15	0.30	**2.32 ^W^**
7	17	0.71	**2.76**	N/A
8	23	1.74	2.59	**2.88 ^w^**
9	30	−0.15	2.55	**2.91 ^B^**

The superscript letter depicts whether the modifications were made to the Weyer et al. primer/probe set (^W^) or the Bachanek-Bankowska et al. primer probe set (^B^). Bold values indicate primers and probes chosen for inclusion in the multiplex PCR-xMAP assay. N/A, not applicable.

**Table 4 microorganisms-12-00932-t004:** Detection of serially diluted AHSV reference strains using multiplex PCR-xMAP methodology and pan-AHSV RT-qPCR. For RT-qPCR, C_T_ values less than 45 were considered positive. For multiplex PCR-xMAP, values above 200 MFI were considered positive. The multiplex PCR-xMAP methodology utilised two separate reactions, a 4-plex reaction (multiplex A) and a 5-plex reaction (multiplex B), to detect all nine AHSV serotypes. ND, not detected.

AHSV Multiplex A					AHSV Multiplex B				
AHSV Serotype	Dilution	RT-qPCR C_T_	Copies/µL	PCR-xMAP MFI	AHSV Serotype	Dilution	RT-qPCR C_T_	Copies/µL	PCR-xMAP MFI
AHSV-1	Neat	24.52	1.81 × 10^4^	5019	AHSV-2	Neat	23.27	4.16 × 10^4^	4312
	10^−1^	28.29	1.56 × 10^3^	1971		10^−1^	26.96	3.73 × 10^3^	3043
	10^−2^	31.88	2.77 × 10^2^	399		10^−2^	31.18	3.58 × 10^2^	1934
	10^−3^	34.73	2.30 × 10^1^	ND		10^−3^	33.49	5.00 × 10^1^	796
	10^−4^	38.26	2	ND		10^−4^	36.82	5	ND
AHSV-3	Neat	22.76	5.73 × 10^4^	3339	AHSV-4	Neat	22.68	4.56 × 10^4^	2216
	10^−1^	26.63	4.53 × 10^3^	3496		10^−1^	25.96	5.39 × 10^3^	1310
	10^−2^	29.92	5.25 × 10^2^	2613		10^−2^	28.85	7.86 × 10^2^	829
	10^−3^	33.65	4.50 × 10^1^	1292		10^−3^	32.43	7.90 × 10^1^	331
	10^−4^	38.25	3	483		10^−4^	36.88	5	ND
AHSV-5	Neat	22.29	5.79 × 10^4^	2349	AHSV-6	Neat	22.20	6.11 × 10^4^	6516
	10^−1^	25.69	6.22 × 10^3^	1936		10^−1^	26.39	3.94 × 10^3^	5542
	10^−2^	29.08	6.76 × 10^2^	1913		10^−2^	29.90	4.03 × 10^2^	4389
	10^−3^	32.27	8.40 × 10^1^	571		10^−3^	33.07	4.90 × 10^1^	2174
	10^−4^	37.50	4	ND		10^−4^	36.64	5	521
AHSV-7	Neat	22.47	5.53 × 10^4^	4358	AHSV-8	Neat	24.26	1.70 × 10^4^	1618
	10^−1^	27.05	2.72 × 10^3^	3195		10^−1^	28.3	1.18 × 10^3^	1293
	10^−2^	30.21	3.36 × 10^2^	1549		10^−2^	31.5	1.38 × 10^2^	878
	10^−3^	33.95	2.90 × 10^1^	540		10^−3^	34.9	1.50 × 10^1^	220
	10^−4^	38.39	3	ND		10^−4^	38.4	1	ND
					AHSV-9	Neat	24.23	1.74 × 10^4^	4227
						10^−1^	27.43	1.61 × 10^3^	4665
						10^−2^	31.49	1.46 × 10^2^	3626
						10^−3^	35.37	1.10 × 10^1^	1953
						10^−4^	38.74	1	320

**Table 5 microorganisms-12-00932-t005:** Summary of diagnostic sensitivity and specificity testing.

Total tested	87		
True positive ^1^	59	False positive	0
False negative ^1^	8	True negative	20
Positive samples correctly serotyped ^2^	59/59		
Serotypes identified	AHSV-1, -2, -4, -5, -6, and -9		

^1^ When assessed against the pan-AHSV RT-qPCR assay. ^2^ When assessed against the serotype-specific RT-qPCR.

## Data Availability

Data are contained within the article.

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
