# Peer review of "Designing a Multiplex PCR-xMAP Assay for the Detection and Differentiation of African Horse Sickness Virus, Serotypes 1–9"

_microorganisms, 2024, doi:10.3390/microorganisms12050932_

Round 1
Reviewer 1 Report
Comments and Suggestions for Authors
This is a well written manuscript reporting significant findings on a high impact transboundary animal disease and I recommend it should be published following minor revision.
Recommendations
Line 29 states "AHSV is endemic in sub-Saharan Africa, but incursions into North Africa4 and non-African...". Reference 4 in the reference list refers to a paper entitled "Review Article West Nile infection of Horses. This reference should be replaced with a more appropriate citation.
Line 53 states "... as some AHSV serotypes are known to cross-react, results can be ambiguous13." Reference 13 in the reference list refers to the EHDV chapter in the WOAH Manual of Diagnostic Tests and Vaccines for Terrestrial Animals. This reference should be to the AHSV chapter in the WOAH manual.
Reference 26 in the reference list is not referred to in the manuscript. It would probably be appropriate to to include it in lines 319 and 321.
Comment/Question for consideration
The SCREENED analysis used to evaluate the primers and probes used in this study provides a very elegant method for in silico evaluation of primers and probes. A total of 224 of the 301 VP2 sequences downloaded from GenBank were used in the SCREENED analysis. Of these, 68 were for AHSV1 and a significant proportion of these VP2 sequences were from AHSV outbreaks associated with reassortment and/or reversion to virulence of vaccine strains in the Western Cape Province of South Africa in 2004, 2011, 2014 and 2016 and as they were all derived from the AHSV1 vaccine strain are therefore not independent. Furthermore, the AHS reference viruses have been sequenced by multiple different laboratories and have been deposited on GenBank and are therefore probably not independent observations. The calculation of an average or mean result usually assumes that the observations are independent. My question is therefore, should identical sequences and or sequences from a common source only have a single representative included in the SCREENED analysis? The authors may wish to consider this and incorporate a comment in the discussion which could assist authors who apply this analysis in the future.
Author Response
We thank you for taking the time to read and review the manuscript. We have done our best to address each comment as stated below.
Line 29 states "AHSV is endemic in sub-Saharan Africa, but incursions into North Africa4 and non-African...". Reference 4 in the reference list refers to a paper entitled "Review Article West Nile infection of Horses. This reference should be replaced with a more appropriate citation.
Thank you for the comment, this has been rectified.
Line 53 states "... as some AHSV serotypes are known to cross-react, results can be ambiguous13." Reference 13 in the reference list refers to the EHDV chapter in the WOAH Manual of Diagnostic Tests and Vaccines for Terrestrial Animals. This reference should be to the AHSV chapter in the WOAH manual.
Thank you for the comment, this has been rectified.
Reference 26 in the reference list is not referred to in the manuscript. It would probably be appropriate to to include it in lines 319 and 321.
Thank you for the comment, this has been rectified.
Comment/Question for consideration
The SCREENED analysis used to evaluate the primers and probes used in this study provides a very elegant method for in silico evaluation of primers and probes. A total of 224 of the 301 VP2 sequences downloaded from GenBank were used in the SCREENED analysis. Of these, 68 were for AHSV1 and a significant proportion of these VP2 sequences were from AHSV outbreaks associated with reassortment and/or reversion to virulence of vaccine strains in the Western Cape Province of South Africa in 2004, 2011, 2014 and 2016 and as they were all derived from the AHSV1 vaccine strain are therefore not independent. Furthermore, the AHS reference viruses have been sequenced by multiple different laboratories and have been deposited on GenBank and are therefore probably not independent observations. The calculation of an average or mean result usually assumes that the observations are independent. My question is therefore, should identical sequences and or sequences from a common source only have a single representative included in the SCREENED analysis? The authors may wish to consider this and incorporate a comment in the discussion which could assist authors who apply this analysis in the future.
Thank you for the comment. We did perform manual inspection of the dataset to try and avoid the incorporation of irrelevant sequences. However, it was not always obvious which sequences should be discounted, and in some cases, it could be subjective. For example, in certain conditions, it may be advisable to include identical sequences. This could occur when you have two identical sequences from different outbreaks, and you may want to weigh the average assay score towards these seemingly more common occurrences. Nevertheless we have taken the comment into consideration and addressed it in the discussion:
“It should be noted that while efforts were made to remove irrelevant VP2 sequences from our dataset, in some cases closely related sequences from a common source were still included. In future work where a similar in silico approach is undertaken, researchers need to carefully consider how including multiple sequences from a single laboratory or outbreak may mean particular mismatches could be over represented in the analysis”
Reviewer 2 Report
Comments and Suggestions for Authors
The paper presents an original variant of PCR assay technique with low detection limit, high diagnostic sensitivity and specificity. The work at whole accords to demands of the Microorganisms journal, but needs some revisions/justifications:
1. The assay should be clearly indicated in the title as amplification (PCR) assay to exclude risks of misunderstanding. The actual title can be applied, for example, to immunoassays.
2. Time of the assay (including samples preparation) is its important parameter that should be specified in the Abstract.
3. Lines 45-47. The difference of AHSVs belonging to different serotypes should be better specified and associated with differences of some indicated genes/proteins. The actual text needs correction: serotypes themselves cannot be based (?) on intermolecular interactions.
4. Considering earlier developments of PCR for AHSVs, the authors indicated refs. [14-17] published between 2000 and 2015. In this situation the necessity to re-evaluate new data is reasonable. However, some multiplex amplification techniques for AHSVs were published after 2015 – see
Development and validation of three triplex real-time RT-PCR assays for typing African horse sickness virus: Utility for disease control and other laboratory applications (2024; DOI 10.3390/v16030470);
Using a new serotype-specific Polymerase Chain Reaction (PCR) and sequencing to differentiate between field and vaccine-derived African horse sickness viruses submitted in 2016/2017 (2019; DOI 10.1016/j.jviromet.2019.01.016).
These works (and maybe, some other works) should be taken into additional consideration for more grounded comments about predecessors.
5. Lines 102-154 give extended description of the implemented in silico screening. Some parts of this text are perceived more as a commented description of the conducted research than as a list of used methods and their parameters. The authors are encouraged to revise this section, consistently describing the methods used, and to move final decision from the results of their in silico analysis to the beginning of the “Results” section.
6. Line 236 and below. The title of this sub-section should contain indication of the implemented techniques for the target detection. The following addressing to Fig. 1 in the text should be transformed to direct sequential indications of the items (i)-(vii) demonstrating each part of the described protocol. It will make the description at lines 237-261 better structured, more visible and clear.
7. Commonly the optimization is the comparative study of the results obtained after varying some parameters with the choices of their best (optimal) values from the studied variants. The list at lines 309-324 presents the proposed decisions that were directly applied to improve the assay protocol. It is not an optimization in the strong sense of this term. I think that the authors' action could be better described as an improvement(s).
8. Legend to Fig. 2. Please indicate the number of experimental repetitions that were used to obtain (calculate) the represented standard deviations. The values for off-target testing are very poorly visible from the Figure. I think that a table with complete set of quantitative MFI data for all target and off-target testings will be useful as a Supplement to the paper.
9. The number of reliable digits should be checked for copies/mL values in Table 4, and the values should be rounded were necessary. For example, 18,143 is probably 18,1 x 10(3) taking into account accuracy of the presented experiments.
10. Similarly, diagnostic specificity and sensitivity cannot be calculated with four reliable digits from data of 67…87 samples testing.
11. The knowledge about individual results of PCR-xMAP testing has a limited scientific importance. So the tables 5, 6 can be moved to the Supplement, and only integral statistics will be reasonable in the core text.
12. The Discussion section gives long (lines 389-502) non-structured comments, and their accordance to specific obtained results is sometimes non-clear and with the necessity of additional efforts for understanding. Some statements repeat common features of amplification techniques and AHSV serotyping that were already known without the consideration of the authors' results. This section should be better structured with clear relation of each conclusion with the presented above experimental data.
Author Response
- The assay should be clearly indicated in the title as amplification (PCR) assay to exclude risks of misunderstanding. The actual title can be applied, for example, to immunoassays.
Thank you for the comment. The title has been changed to make it clear that the assay includes PCR:
Title changed to: Designing a multiplex PCR-xMAP assay for the detection and differentiation of African horse sickness virus serotypes 1-9
- Time of the assay (including samples preparation) is its important parameter that should be specified in the Abstract .
Thank you, the timings have now been added to the abstract (line 16-17) and discussion (line 511-512)
- Lines 45-47. The difference of AHSVs belonging to different serotypes should be better specified and associated with differences of some indicated genes/proteins. The actual text needs correction: serotypes themselves cannot be based (?) on intermolecular interactions.
Thanks for the comment. These line have been re-worded to try and improve their meaning and add some detail.
“As one of the two outermost proteins of the AHSV capsid, the VP2 spike, which is encoded by genome segment 2 and is comprised of four domains (hub, body, hairpin and external tip), is a major target of neutralising antibodies. This has driven selective pressure on the encoding gene that has resulted in it being the least conserved of the AHSV proteins12. The neutralizing antibodies generated against VP2 are serotype-specific. Nine serotypes of AHSV have been identified based on the ability of serotype-specific antibodies to neutralise the virus. 12,13. “
- Considering earlier developments of PCR for AHSVs, the authors indicated refs. [14-17] published between 2000 and 2015. In this situation the necessity to re-evaluate new data is reasonable. However, some multiplex amplification techniques for AHSVs were published after 2015 – see
Development and validation of three triplex real-time RT-PCR assays for typing African horse sickness virus: Utility for disease control and other laboratory applications (2024; DOI 10.3390/v16030470);
Using a new serotype-specific Polymerase Chain Reaction (PCR) and sequencing to differentiate between field and vaccine-derived African horse sickness viruses submitted in 2016/2017 (2019; DOI 10.1016/j.jviromet.2019.01.016).
These works (and maybe, some other works) should be taken into additional consideration for more grounded comments about predecessors.
Thank you for the suggestion and a re-review of current literature has been done and the publications suggested above have been incorporated.
- Lines 102-154 give extended description of the implemented in silico screening. Some parts of this text are perceived more as a commented description of the conducted research than as a list of used methods and their parameters. The authors are encouraged to revise this section, consistently describing the methods used, and to move final decision from the results of their in silico analysis to the beginning of the “Results” section.
Thank you for your suggestion. We felt that it was necessary to include an illustrative example of our scoring criteria in the in silico methods section to clarify the approach used. We feel that this is necessary to assist the reader in understanding the methodology, however we agree that this could be better clarified in the text. Accordingly, we have adjusted the text to emphasise that this serves solely as an illustrative example of the approach used (line 135). Furthermore, since it represents only one instance among a vast set of data analysed, it wouldn't be appropriate for inclusion in the results section (where we provide a summary of the results).
- Line 236 and below. The title of this sub-section should contain indication of the implemented techniques for the target detection. The following addressing to Fig. 1 in the text should be transformed to direct sequential indications of the items (i)-(vii) demonstrating each part of the described protocol. It will make the description at lines 237-261 better structured, more visible and clear.
Thank you for the suggestion. We have altered the Title of the subsection changed to “xMAP based detection of serotype specific amplicons”. Sequential indicators aligning with figure 1 have also been added to the manuscript as suggested.
- Commonly the optimization is the comparative study of the results obtained after varying some parameters with the choices of their best (optimal) values from the studied variants. The list at lines 309-324 presents the proposed decisions that were directly applied to improve the assay protocol. It is not an optimization in the strong sense of this term. I think that the authors' action could be better described as an improvement(s).
Thank you for your comment. The development of the assay did include the comparative study of the results obtained after varying parameters. Data from these experiments has now been collated and summarised in the supplementary data. We presented the list on lines 319-334 to summarize the results of the optimization. The sentence on lines 316-318 has been changed to try and reflect how this .
"A series of different assay parameters were assessed using a range of conditions to monitor their effect on assay performance. Based on the results of these optimization experiments, we decided upon the following approaches"
- Legend to Fig. 2. Please indicate the number of experimental repetitions that were used to obtain (calculate) the represented standard deviations. The values for off-target testing are very poorly visible from the Figure. I think that a table with complete set of quantitative MFI data for all target and off-target testings will be useful as a Supplement to the paper.
Thank you for the comment. A full data set has now been provided in the supplementary data as suggested. The number of repetitions has been added to the figure legend.
- The number of reliable digits should be checked for copies/mL values in Table 4, and the values should be rounded were necessary. For example, 18,143 is probably 18,1 x 10(3) taking into account accuracy of the presented experiments.
Copies / uL values have been checked and where appropriate rounded as suggested.
- Similarly, diagnostic specificity and sensitivity cannot be calculated with four reliable digits from data of 67…87 samples testing.
Numbers for displaying diagnostic specificity and sensitivity have been converted to integers as suggested.
- The knowledge about individual results of PCR-xMAP testing has a limited scientific importance. So the tables 5, 6 can be moved to the Supplement, and only integral statistics will be reasonable in the core text.
Thank you for the suggestion. Table 5 and 6 have been moved to supplementary data and a new table (Table 5) created for inclusion in the main text. This table only includes the integral statistics.
- The Discussion section gives long (lines 389-502) non-structured comments, and their accordance to specific obtained results is sometimes non-clear and with the necessity of additional efforts for understanding. Some statements repeat common features of amplification techniques and AHSV serotyping that were already known without the consideration of the authors' results. This section should be better structured with clear relation of each conclusion with the presented above experimental data.
Thank you for the comment. We have revised the discussions to try and clarify which sections of the text correspond with specific results. Additionally, we have added more detail in certain areas and streamlined the text in others, which we hope makes the overall discussion more structured.
Round 2
Reviewer 2 Report
Comments and Suggestions for Authors
The manuscript has been successfully revised and now may be published